# The role of interfacial donor–acceptor percolation in efficient and stable all-polymer solar cells

Zhen Wang [1], Yu Guo [1], Xianzhao Liu [1], Wenchao Shu[2], Guangchao Han[2], Kan Ding[3], Subhrangsu Mukherjee[3], Nan Zhang[4], Hin-Lap Yip [4,5,6], Yuanping Yi [2], Harald Ade [3] & Philip C. Y. Chow [1] ✉

Polymerization of Y6-type acceptor molecules leads to bulk-heterojunction organic solar cells with both high power-conversion efficiency and device stability, but the underlying mechanism remains unclear. Here we show that the exciton recombination dynamics of polymerized Y6-type acceptors (Y6-PAs) strongly depends on the degree of aggregation. While the fast exciton recombination rate in aggregated Y6-PA competes with electron-hole separation at the donor–acceptor (D–A) interface, the much-suppressed exciton recombination rate in dispersed Y6-PA is sufficient to allow efficient free charge generation. Indeed, our experimental results and theoretical simulations reveal that Y6-PAs have larger miscibility with the donor polymer than Y6-type small molecular acceptors, leading to D–A percolation that effectively prevents the formation of Y6-PA aggregates at the interface. Besides enabling high charge generation efficiency, the interfacial D–A percolation also improves the thermodynamic stability of the blend morphology, as evident by the reduced device "burn-in" loss upon solar illumination.

Organic solar cells (OSCs) based on synthetic molecules and polymers are promising candidates for low-cost and flexible photovoltaic (PV) panels that can be seamlessly integrated with our environment[1]. Over the past few years, the development of non-fullerene, Y6-type small-molecule acceptors (Y6-SMAs) have greatly improved the solar power conversion efficiencies (PCEs) of OSC devices, now reaching ~19% in single-junction solar cells which are comparable to those of commercial inorganic solar cells[2–4]. Scaling up the fabrication of OSC devices using layer-by-layer and other printing methods has also made significant progress, paving the way towards large-scale application of OSCs[5–9]. However, today's OSC devices based on Y6-SMAs (which are blended with donor polymers such as PM6) generally suffer from fast PV performance degradation under solar irradiation, with $T_{80}$ lifetimes

typically no more than ~500–1000 h ($T_{80}$ lifetime marks the time at which PCE drops to 80% of its initial value)[10]. OSC devices based on other types of non-fullerene SMAs (e.g., IT-4F and IEICO-4F) as well as polymeric acceptors (i.e., N2200)[11] show longer device lifetimes, but they generally show lower PCEs (~8–14%)[12,13].

The limited device operation lifetime of state-of-the-art OSCs based on Y6-SMAs is attributed to the instability of the nanoscale bulk-heterojunction (BHJ) morphology that is optimized for charge photogeneration[14,15]. Due to the hypo-miscible nature of the blend, its deviation from the efficiency-optimized morphology during solar PV operation is driven by (i) thermodynamic relaxation of the mixed domains in the blend from an initially quenched composition to the binodal, and (ii) kinetic diffusion of the small-molecule acceptors and

[1]Department of Mechanical Engineering, The University of Hong Kong, Pokfulam, Hong Kong SAR, China. [2]CAS Key Laboratory of Organic Solids, Institute of Chemistry, Chinese Academy of Sciences, Haidian, Beijing 100190, China. [3]Department of Physics and Organic and Carbon Electronics Laboratories (ORaCEL), North Carolina State University, Raleigh, NC 27695, USA. [4]Department of Materials Science and Engineering, City University of Hong Kong, Kowloon, Hong Kong SAR, China. [5]School of Energy and Environment, City University of Hong Kong, Kowloon, Hong Kong SAR, China. [6]Hong Kong Institute for Clean Energy, City University of Hong Kong, Kowloon, Hong Kong SAR, China. ✉e-mail: pcyc@hku.hk

polymer donors that causes nucleation and phase separation[14,15]. Recently, researchers have successfully developed polymerized-Y6-type acceptors (Y6-PAs) for all-polymer OSCs that can achieve decent PCE values (-15–18%), some also show superior device stabilities compared to their Y6-SMA counterparts (see Supplementary Table 1)[16–23]. A recent study by Huang and co-workers showed that, with increasing degree of polymerization, Y6-PA systems show reduced diffusion coefficient compared to the Y6-SMAs. This in turn leads to a kinetic stabilization of the blend morphology that improves the device lifetime (with remarkable $T_{80}$ lifetimes up to -25000 h)[24]. It is thus clear that all-polymer OSCs based on Y6-PAs are highly promising material candidates for efficient and stable OSCs. However, while the photophysical mechanism for charge photogeneration at the donor–acceptor (D–A) interface have been well-studied for Y6-SMA-based OSC blends, a fundamental understanding of charge generation process for efficient all-polymer OSC blends is still lacking. Furthermore, the precise morphology at the all-polymer D–A interface, as well as its role in charge photogeneration and morphological stability, has remained unclear.

Previous studies have highlighted the roles of local exciton–charge transfer (CT) state hybridization[25,26], long exciton diffusion length[27,28], electrostatic interactions at the interface[29,30], electronic delocalization[31], suppression of spin-triplet recombination[32,33], intra-moiety excimers[34], direct free carrier generation[35,36], among other properties in enabling high PCEs in Y6-SMA-based OSCs. Due to the small D–A energetic offset, the dissociation of excitons and CT states into free carriers at the D–A interface is an endothermic process that takes up to -100 ps at room temperature[37]. As such, during PV operation at thermal equilibrium, a nanosecond exciton lifetime is needed to enable high charge separation yields. This is well described by Brabec and co-workers using a Boltzmann model and helps to explain why near-infrared-absorbing SMA materials with short exciton lifetimes show poor charge photogeneration yields[38].

Here we investigate the interfacial charge photogeneration mechanism and excited state dynamics in three model Y6-PA-based all-polymer OSC systems (namely, PM6:PY-IT[39], PM6:PYF-T-o[18,40], and PM6:PY-V-γ[41]). Transient absorption (TA) spectroscopy measurements reveal that free carrier generation in these all-polymer blends takes -100 ps to complete. This is largely comparable to the charge generation timescale found in PM6:Y6-SMA blends, thus indicating that a long (nanosecond) exciton lifetime is also needed for efficient charge photogeneration in Y6-PA-based blends. Nevertheless, we observe significantly shorter exciton lifetimes in the neat Y6-PA films (-0.3–0.5 ns), which seemingly contradicts their high charge photogeneration efficiencies. Interestingly, we find that exciton lifetimes in Y6-PAs are strongly dependent on the degree of molecular aggregation, with dispersed Y6-PAs (either in solution or in polymer matrix) showing significantly extended exciton lifetimes (-1.0–1.9 ns). Based on a combination of molecular dynamics (MD) simulations and morphological/compositional characterization results, we propose that large D–A miscibility in these all-polymer blends leads to the formation of dispersed and stretched Y6-PAs at the D–A interface. The nanosecond exciton lifetime of the dispersed and stretched Y6-PAs is significantly longer than the interfacial charge separation time, thereby enabling high charge generation yields at thermal equilibrium. Furthermore, we show that the large miscibility of the all-polymer blends enhances the thermodynamic stability of the blend morphology compared to Y6-SMA-based blends.

## Results

### Basic optical properties and frontier molecular orbitals

Figure 1a shows the chemical structures of Y6 and the three Y6-PAs studied herein. The polymerized acceptors have 6–8 repeating units, which was previously found to enable optimal performance in blend devices[18,39,41]. Fig. 1b, c show their absorption and photoluminescence (PL) spectra in chloroform (CF) solutions and in neat films,

respectively. A comparison of the absorption peaks in solution and film shows a larger difference in Y6 (-100 nm) than in the three Y6-PAs (-20 nm). A similar trend is also found for the PL data. Strong intermolecular electronic couplings have been assigned as the reason for the large absorption/PL shifts observed in Y6 from solution to film[31]. The much-reduced spectral shifts observed for Y6-PA systems suggest that the excited state properties are dominated by intra-chain electron couplings. We perform density functional theory (DFT) calculations to study the frontier molecular orbital distributions of isolated Y6 monomer and PY-IT chain (see Fig. 1d, e; corresponding results for PYF-T-o and PY-V-γ can be found in Supplementary Fig. 1). While both the highest-occupied (HOMO) and lowest-unoccupied (LUMO) molecular orbitals of Y6 monomer are confined to the molecular backbone, we find that the LUMO of Y6-PA systems are significantly extended (-2–3 units) along the polymer chain. Note that HOMO-1 is shown for PY-IT since the HOMO-1 to LUMO transition represents the configuration with the highest contribution to the transition moment of the $S_0$-$S_1$ transition (see Supplementary Fig. 1). Such intra-chain wavefunction delocalization helps to explain the lower optical gap of isolated Y6-PA chains compared to isolated Y6 molecules in dilute solutions. We also use time-dependent DFT to calculate the absorption spectra and oscillator strengths of the Y6 monomer and the three Y6-PA chains (Supplementary Fig. 2), which are in good agreement with the experimental results.

### Photoexcited state properties and dynamics in neat and blended films

We use TA spectroscopy to investigate the photoexcited state dynamics in these acceptors and their D–A blends with polymer donor PM6 (Supplementary Fig. 3). Figure 2a, b show the TA data of PM6:Y6 and PM6:PY-IT blends probed between -500–910 nm. By pumping at 750 nm, we selectively photoexcite the acceptor molecules/polymers. Note that additional TA data for neat and blended films covering -500–1600 nm are found in Supplementary Figs. 4–10 for reference. At early times (<1 ps), we find that the acceptor ground state bleaching (GSB) signals dominate the TA response for all four blends (-800–870 nm). With increasing time delay, we observe: (1) a signal growth between -550–650 nm that coincides with the GSB feature of PM6, and (2) a negative signal growing between -670–800 nm that can be attributed to polaron absorption of PM6[32,34,42]. Both features are signatures of hole transfer of the photoexcited excitons at the D–A interface that subsequently lead to the formation of CT states and free carriers. For all four D–A blends, we find that the donor polymer GSB and polaron absorption signals are maximized at -50–100 ps. The same observation is found when the pump energy is set to 550 nm (exciting both donor and acceptor materials), as shown in Supplementary Fig. 11–12. This result indicates that charge generation dynamics in PM6:Y6 and PM6:Y6-PA blends are similar, taking -100 ps to complete, and therefore a long (nanosecond) exciton lifetime in the low-gap acceptor material is needed to promote efficient charge generation[37,38]. Note that although the HOMO offsets estimated from cyclic voltammetry method for the all-polymer blends are slightly larger than that of PM6:Y6 (-0.12–0.16 eV compared to 0.09 eV)[18,39,41], all of these systems fall into the charge-transfer-state–exciton equilibrium regime in which a long exciton lifetime is needed for efficient charge generation[38].

We then study the acceptor exciton lifetimes using time-resolved photoluminescence (TRPL). As shown in Fig. 2c, in neat film, Y6 exhibits a slow PL decay and an exciton lifetime of -1.36 ns, whilst PY-IT, PYF-T-o and PY-V-γ neat films show much faster decays and consequently shorter exciton lifetimes of 0.39, 0.43 and 0.50 ns, respectively. The short exciton lifetimes of Y6-PAs are inconsistent with their decent device performance and seemingly contradict the aforementioned notion that long exciton lifetime is needed for efficient charge generation at the D–A interfaces[38]. However, we find much longer exciton lifetimes in dilute solutions (0.03 mg mL$^{-1}$ in CF) of PY-IT, PYF-T-o and PY-V-γ compared to

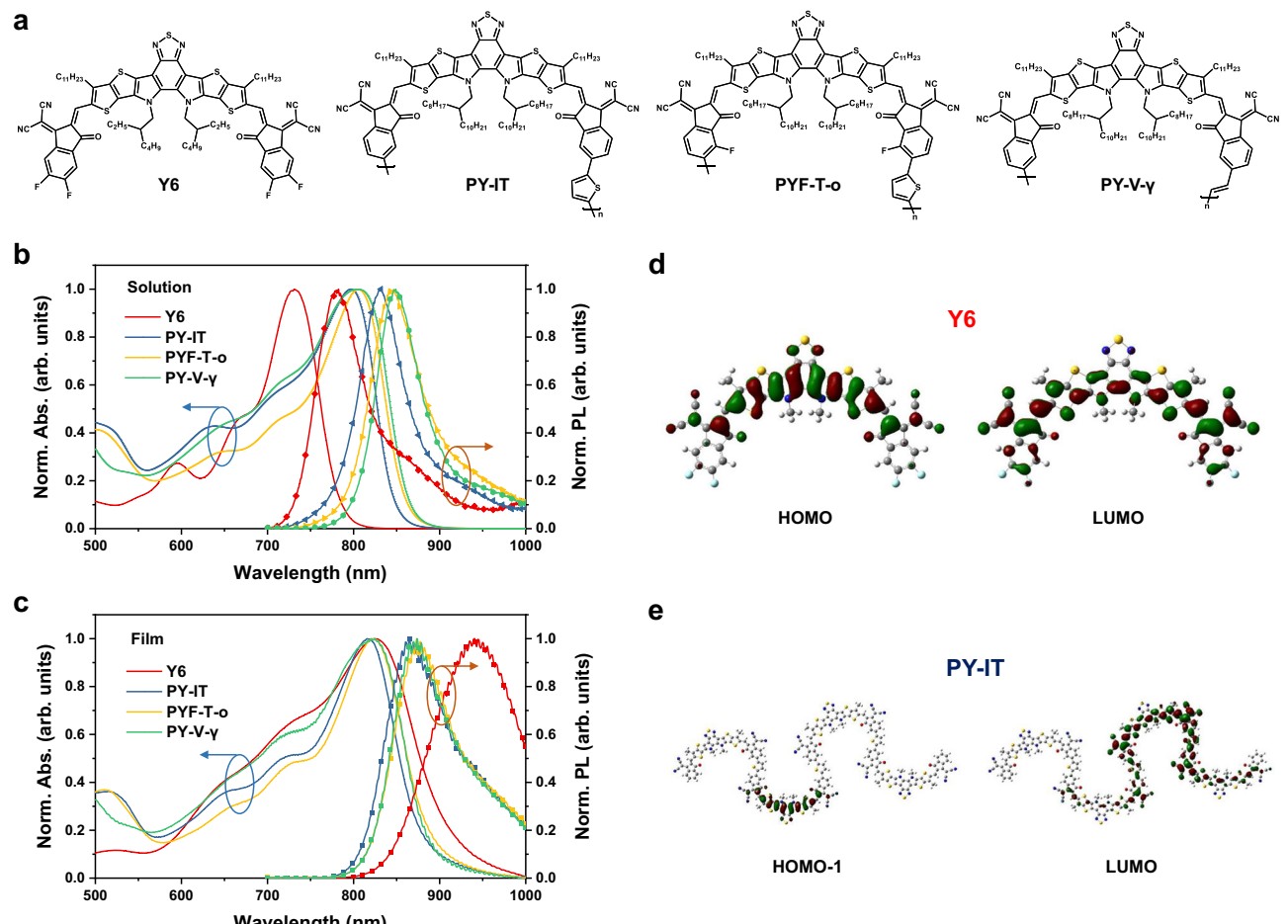

**Fig. 1 | Molecular structures and basic properties. a** Molecular structures of Y6, PY-IT, PYF-T-o and PY-V-γ. **b** Normalized absorption spectra and photo-luminescence (PL) spectra (excited at 680 nm) of Y6, PY-IT, PYF-T-o and PY-V-γ solutions in chloroform. **c** Normalized absorption spectra and PL spectra (excited at 720 nm) of Y6, PY-IT, PYF-T-o and PY-V-γ films. **d, e** Frontier orbitals of Y6 (**d**) and PY-IT (**e**) with the highest contribution to the transition moment of the $S_0$-$S_1$ transition, where side chains are substituted by methyl group. Similar results are found for PYF-T-o and PY-V-γ, see Supplementary Information.

films (decay lifetimes of 1.89, 1.03, and 1.83 ns, respectively, see Supplementary Figs. 13, 14), thus indicating that exciton lifetime is strongly dependent on the degree of aggregation. Concentration-dependent absorbance/PL data confirm that the degree of aggregation is negligible for the dilute Y6-PA solutions (see Supplementary Figs. 15–21 and Table 2). We further study the aggregation effects in solid state by measuring TRPL of the Y6-PAs dispersed in an insulating polymer matrix of polyvinylcarbazole (PVK)[43] with various weight fractions. Similar to the solution data, we observe much increased exciton lifetimes in Y6-PAs when the dilution weight fraction falls below ~5% (see Fig. 2d, e and Supplementary Fig. 22 and Table 3), reaching ~1.4 ns at 1% weight fraction. The PL quantum yields (PLQYs) of neat PY-IT, PYF-T-o and PY-V-γ films are measured and estimated to be 1.8%, 1.4% and 1.7% respectively, which is lower than that of Y6 films (~3.5%). Consistent with the extended decay lifetime, we measure much-increased PLQYs in dilute solutions (31.3%, 14.0% and 19.3% respectively). We determine the radiative ($\kappa_r$) and non-radiative recombination rates ($\kappa_{nr}$) according to the following expressions of exciton lifetime ($\tau$) and PLQY ($\eta$)[44], see Supplementary Fig. 23 and Table 4.

$$\tau = \frac{1}{\kappa_r + \kappa_{nr}} \qquad (1)$$

$$\eta = \frac{\kappa_r}{\kappa_r + \kappa_{nr}} \qquad (2)$$

We find little difference between $\kappa_r$ of Y6 and the three Y6-PAs in both films and solutions (~3 × 10⁷ s⁻¹ in films and ~2 × 10⁸ s⁻¹ in solutions). While $\kappa_{nr}$ of solution samples are also similar between these acceptors in solutions (~5 × 10⁸ s⁻¹), we find that $\kappa_{nr}$ values of the three Y6-PAs in neat films are an order of magnitude greater compared to Y6 neat films (~2.2 × 10⁹ s⁻¹ compared to ~2.0 × 10⁸ s⁻¹). This confirms that the short exciton lifetime of neat Y6-PA films is due to fast non-radiative recombination in aggregated (film) state. Such aggregation-induced non-radiative recombination is also observed in many other molecular and polymeric materials[45–48]. These optical characterization results imply that aggregation effects in neat films greatly reduce the exciton lifetimes of PY-IT, PYF-T-o and PY-V-γ compared to their iso-lated state in dispersed solutions/films, leading to fast non-radiative recombination that would compete with charge generation at the D–A interface[38]. Since exciton lifetime is much extended in dispersed Y6-PA chains, we speculate that the interplay of aggregation effects and exciton lifetime plays an important role in the charge generation process at the D–A interface. The PL spectra of the dispersed Y6-PA films are blue-shifted with respect to neat films (Supplementary Fig. 24). When these Y6-PAs are blended with PM6, we find that the resulted PL spectra is also slightly blue-shifted with respect to the neat acceptor films (Supplementary Fig. 25), which supports the notion that the Y6-PAs are partially dispersed in D–A blends. We also performed TA and TRPL for the PY-monomer, which has the same side chains as Y6-PAs studied in this work (see Supplementary Fig. 26–29). We observe little difference in exciton dynamics between the PY-

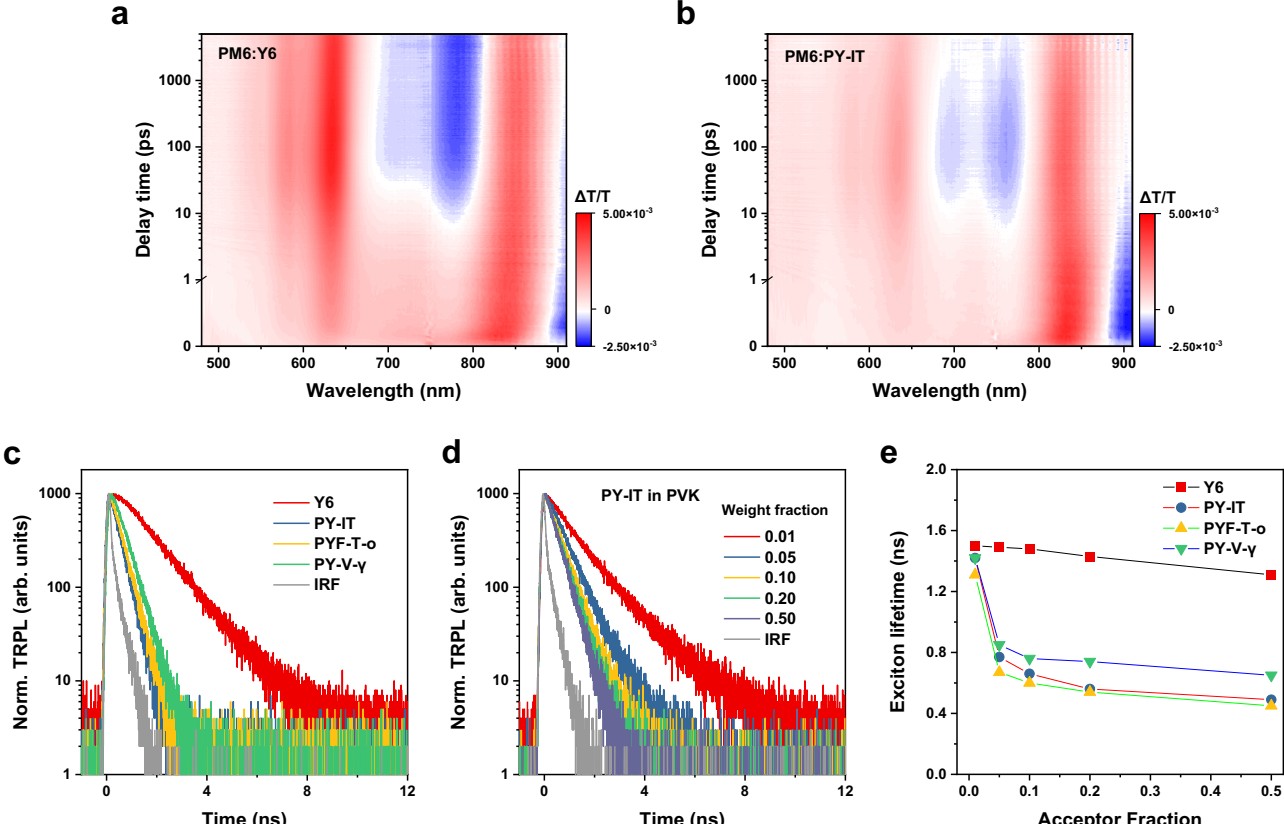

**Fig. 2 | Time-resolved optical properties. a, b** Transient absorption (TA) spectroscopy results for PM6:Y6 (**a**) and PM6:PY-IT (**b**) blended films, both excited at 750 nm. Additional TA results including data for pristine films and blends in the near-infrared regions can be found in the Supplementary Information. **c** Time-resolved photoluminescence (TRPL) data of Y6, PY-IT, PYF-T-o and PY-V-γ films, all excited at 720 nm. **d** TRPL data of PY-IT dispersed in polyvinylcarbazole (PVK) with various acceptor weight fractions, all excited at 680 nm. Similar results for PYF-T-o and PY-V-γ are found in Supplementary Information. **e** Exciton lifetime comparison on PVK-dispersed acceptors with various acceptor weight fractions.

monomer and Y6 films, which implies that the changes in exciton dynamics observed in the Y6-PAs are results of polymerization rather than the side chains.

## Molecular interactions and conformations

Since exciton recombination dynamics of Y6-PAs greatly depends on their degree of aggregation, we perform atomistic MD simulations to study the molecular configurations and intermolecular interactions. Figure 3a, b show the simulated equilibrated snapshots and counts of contacting D−A pairs in PM6:Y6 and PM6:PY-IT blends, simulated using a solvent-evaporating process (see Supplementary Figs. 30−S34 and Note 1 for details)[49,50]. The total counts for contacting D−A pairs, which is defined as the number of D−A pairs with over 6 contacting atoms, are 1191 and 1178 for PM6:Y6 and PM6:PY-IT, respectively. Accounting for the difference in molecular weights between Y6 and PY-IT, the average numbers of contacting D−A pairs are found to be 2.68 and 3.27 for PM6:Y6 and PM6:PY-IT, respectively. This implies that, with same D−A weight ratio, the polymeric PY-IT acceptors can establish more connections with the donor PM6 compared to the Y6 molecular acceptors. This translates to a better miscibility between PM6 and PY-IT compared to PM6 and Y6.

Furthermore, the average end-to-end distance and radius of gyration ($R_g$) of PM6 chains in both PM6:Y6 and PM6:PY-IT blends are calculated to evaluate their conformations. As shown in Fig. 3c, d, the average end-to-end distances of PM6 chains blended with PY-IT (11.00 nm in solution and 8.62 nm in film) are shorter than those in the PM6:Y6 blend (11.49 nm in solution and 9.01 nm in film). The calculated $R_g$ of PM6 chains in PM6:PY-IT blend (3.79 nm in solution and 3.15 nm in film) are also shorter than those in PM6:Y6 blend (3.94 nm in

solution and 3.26 nm in film), see Supplementary Fig. 35. (Note that $R_g$ is defined as $R_g = \sqrt{\frac{\sum_i m_i \|r_i\|^2}{\sum_i m_i}}$, where $m$ is the mass of the atoms and $r$ is the distance of the atoms to the mass center of the chain.) These results indicate that PM6 chains in PM6:Y6 blend is more stretched, while in the PM6:Y6-PA blends they are more twisted/curved. Thus, the PM6 chains are expected to be more entangled with Y6-PAs at the interface. A comparison of the conformation of PY-IT chains is also made between PM6:PY-IT blend and PY-IT neat phase. In both solution and film state, PY-IT chains in PM6:PY-IT blend show larger end-to-end distances than those in neat phase (Supplementary Fig. 36), indicating more stretched conformation of PY-IT chains when blended with PM6 than in neat phase. These MD simulation results imply that the miscibility between PM6 and Y6-PA is greater than PM6 and Y6, leading to more intermixing and the formation of dispersed and stretched Y6-PA chains at the D−A interface.

## Blend morphology and D−A miscibility

We then turn to X-ray scattering experiments to study the morphology. We first perform grazing-incidence wide-angle X-ray scattering (GIWAXS) characterization for both neat and blended films. The results are displayed in Supplementary Fig. 37−39 and Table 5. Overall, we find evidence that Y6-PAs are less ordered than Y6 in both neat and blended films. To further understand the in-plane morphology and composition information, we perform resonant soft X-ray scattering (R-SoXS) characterization[51]. Fig. 4a, b show the Lorentz-corrected thickness-normalized R-SoXS profiles of the neat polymer films and PM6:acceptor blended films, all acquired at 284 eV (see Supplementary Fig. 40, 41 for the raw R-SoXS profiles). It is noted that all films

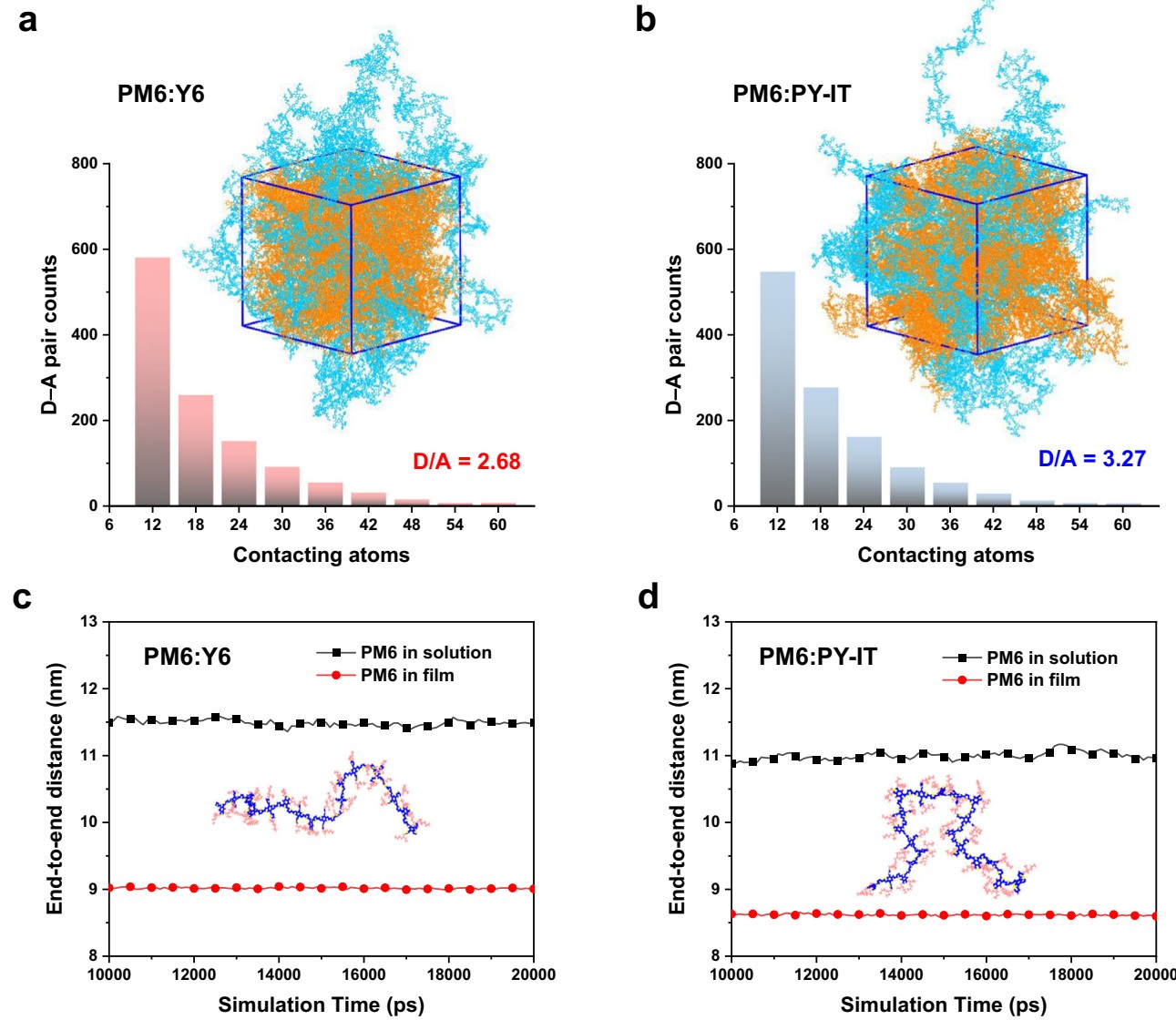

**Fig. 3 | Molecular dynamics (MD) simulations. a, b** MD simulated equilibrated snapshots and counts of contacting donor–acceptor (D–A) pairs for (**a**) PM6:Y6 and (**b**) PM6:PY-IT blends. Details of the simulation process are found in Methods and Supplementary Information. Here we define two atoms are in contact when their separation distance is smaller than the sum of their van der Waals radii, with more contacting counts indicating larger D–A molecular interactions. **c, d** Calculated end-to-end distances as a function of equilibration time of PM6 chains (both in solution and film state) in PM6:Y6 (**c**) and PM6:PY-IT (**d**) blends. "End-to-end distance" is defined as the distance between the two atoms at the end of the polymer chain backbone, with straight chains giving the largest end-to-end-distance.

show R-SoXS features over a range of q vectors, which correspond to a distribution of long periods that is related to the domain sizes (i.e., hierarchical morphologies). As shown in Fig. 4b, PM6:Y6 film exhibits broader scattering features ($0.02–0.2$ nm$^{-1}$) than that of the three PM6:Y6-PA films ($0.02–0.06$ nm$^{-1}$), indicating that the all-polymer blends have, on average, larger domain sizes. To extract information about the phase composition and domain purity, we compare the root-mean-square of the integrated area underneath the R-SoXS profiles (i.e., total scattering intensity)[52]. By normalizing to the data for PM6:Y6 film (taking it as 1), we find that the root-mean-square composition variations for PM6:PY-IT, PM6:PYF-T-o and PM6:PY-V-γ films are 0.77, 0.72 and 0.77, respectively. This result indicates that the PM6:Y6 blend has higher domain purity than the three PM6:Y6-PA blends. Crucially, the result implies that the three Y6-PAs have larger miscibility with the donor PM6 (compared to Y6-SMA), which is in close agreement with our MD simulation results. We also perform optical and morphology characterizations for another donor polymer (D18-Cl) blended with Y6 and the Y6-PAs (see Supplementary Fig. 42–45). Note that high device efficiencies are achieved in these D18-Cl-based blends ($-17–18\%$)[53,54]. Similar to the PM6 blends, in R-SoXS results the root-mean-square composition variations for D18-Cl:PY-IT, D18-Cl:PYF-T-o, and D18-Cl:PY-V-γ are 0.84, 0.86 and 0.86 respectively after normalization to the data of D18-Cl:Y6.

## Discussion

A quantitative understanding of the D–A miscibility in these blended systems can be obtained from the MD simulation and R-SoXS results. We find that the molecular interactions (D–A contacting) of PM6–PY-IT is over 22% higher than that of PM6–Y6, and the relative domain purities of PM6:Y6-PA blends are only $-72–77\%$ of that of PM6:Y6 blend ($-84–86\%$ for D18-Cl-based blends). Therefore, it is reasonable to conclude that Y6-PAs have larger miscibility with these high-performance donor polymers than Y6-SMAs. Schematic illustrations of the nanoscale D–A interfacial percolation in the Y6-SMA and Y6-PA blends are shown in Fig. 4c. We note that the blend morphology of efficient Y6-PA-based blends is similar to the morphology of traditional

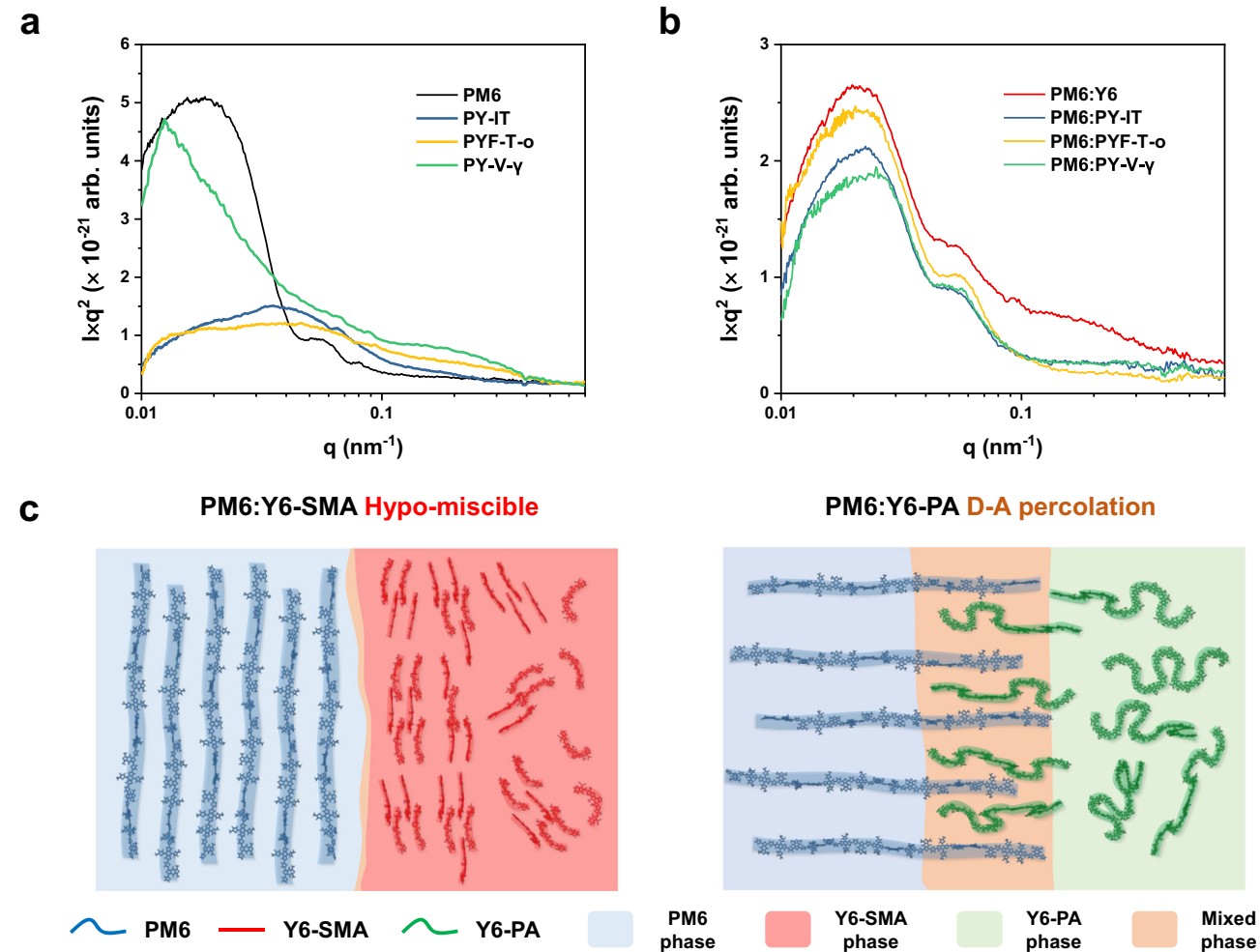

**Fig. 4 | Morphology and miscibility. a**, **b** Lorentz-corrected thickness-normalized resonant soft X-ray scattering (R-SoXS) profiles of neat (**a**) and blended (**b**) films, acquired at 284 eV. Fluorescence signals at high q were subtracted. Here, q values are corresponding to the long periods (L) with a relation of L = 2π/q, and the integrated area underneath the profiles (total scattering intensity) reflects the domain purity of the blends. **c** A schematic illustration of the D–A interfacial percolation of Y6-PA to PM6, in comparison to PM6:Y6-SMA.

fullerene-based blends with three phases[55], which is also generally more stable than the non-fullerene SMA-based devices (hypo-miscible)[15]. Furthermore, a comparison of the estimated binodal profiles of PM6:Y6-SMA and PM6:Y6-PA blends from the Flory–Huggins free energy of mixing equation for polymer solutions also supports our conclusion (see Supplemental Note 2 and Fig. 46)[56]. The larger D–A miscibility in PM6:Y6-PA systems compared to PM6:Y6-SMA systems means that PM6:Y6-PA blends are closer to the percolation threshold compared to the PM6:Y6-SMA blend[57]. As mentioned above, there are mainly two ways for OSC morphological degradation, which involve thermodynamic (miscibility) and kinetic (diffusion) factors. According to the interaction–diffusion framework developed by Ghasemi and co-workers[15], D–A blends with decent percolations and low molecular diffusion coefficients show much better morphological stability. Recent studies have highlighted the importance of suppressing molecular diffusion by oligomerization/polymerization of the Y6-type acceptors[24] and increasing the glass transition temperature ($T_g$)[58] or cold crystallization temperature ($T_{cc}$)[59]. Our results suggest that, in addition to suppressing molecular diffusion, polymerization of Y6-type acceptors could also enhance the thermodynamic stability of the blend morphology via the increase in D–A miscibility and percolation at the interface.

To study the structure–device property relationship, we fabricate device samples of PM6:Y6 and the three PM6:Y6-PAs blends and monitor the change in their PV performance under continuous solar illumination (see Supplementary Fig. 47 and Table 6). The reduced domain purity that we revealed for the Y6-PA blends is consistent with their faster carrier decay (Fig. 2c) and lower device PCE (~13.2–15.5%) compared to Y6 blend (~16.0%). Nevertheless, these Y6-PA blends are still among the most efficient all-polymer OPV systems to date, and we consider that high intra-chain electron transport in these materials helps to reduce recombination losses by allowing charges to move away from the mixed D–A interface[60]. Engineering of the intra-chain charge transport for both polymer donor and acceptor materials may provide a design pathway to further suppress recombination losses in all-polymer blends. For device stability, it is clearly shown that PM6:Y6 device suffered from a more rapid decay in efficiency upon solar illumination compared to the three PM6:Y6-PA blends. This rapid efficiency drop is attributed to the device "burn-in", which is often observed in non-fullerene SMA-based blends with hypo-miscible D–A morphology (over-purification) at the interfaces[61]. Thanks to the increased D–A miscibility and interfacial percolation in the Y6-PA blends, these all-polymer blends suffer from reduced "burn-in" losses and thus an improved device stability over their SMA counterpart. We consider that the increased D–A miscibility in Y6-PA-based blends helps to explain the overall improvement in device stability compared to Y6-SMA-based blends (see Supplementary Note 3 for extended discussion). Nevertheless, it should be noted that besides blend

morphology stability, other factors such as polymer donor degradation[62] and interfacial defects[63] are also likely to affect the overall device stability. Therefore, in addition to developing strategies to improve morphological stability, future research should also target to overcome these factors in order to achieve highly stable and efficient OPV devices that can meet industry requirements.

In conclusion, our results shed light on the fundamental structure–property relationship of state-of-the-art all-polymer OSC systems based on Y6-PAs. TA spectroscopy results show that interfacial charge separation in PM6:Y6-PA blends occurs on a similar timescale as in PM6:Y6-SMA blends (~100 ps), thus indicating that excitons in the low-gap Y6-PAs at the D–A interface should have relatively long lifetime (~1 ns) to achieve efficient charge generation. While aggregates of Y6-PA in neat films suffer from short exciton lifetimes (~0.3–0.5 ns), significantly extended exciton lifetimes are found in dispersed Y6-PA chains (~1.1–1.9 ns), thus indicating that dispersed Y6-PA chains present at the D–A interface are responsible for facilitating charge generation. This is well supported by MD simulations and R-SoXS data that show increased miscibility between PM6 and Y6-PA (compared to Y6-SMA). The increased intermixing leads to dispersed and stretched Y6-PA chains into the PM6 polymer chains, and such D–A percolation at the interface plays a key role in enabling efficient charge generation at thermal equilibrium. Furthermore, the large D–A miscibility in the Y6-PA systems leads to a blend morphology with better thermodynamic stability. Our results reveal the important roles of interfacial D–A percolation in enabling efficient and stable all-polymer solar cells, providing design guidelines for future material and device development.

## Methods

### Materials and sample preparation
PM6, D18-Cl, Y6, PYF-T-o, PY-V-γ and PY-monomer were supplied by eFlexPV Limited. PY-IT was supplied by Hyper Chemical Incorporation. Film samples are spin-cast (2000 rpm) onto clean glass substrates (1 × 1 inch$^2$) from CF solutions, with concentrations of 5 mg mL$^{-1}$ for polymers and 10 mg mL$^{-1}$ for Y6, in a nitrogen-atmosphere glovebox. Then another piece of glass is attached to each sample and the edges are sealed with UV curing epoxy for encapsulation. Solution samples are diluted from 1 mg mL$^{-1}$ solution with CF in glovebox and transferred to a quartz cuvette before measurements. All chemicals were used without further purifications.

### Optical characterizations
UV-vis absorption spectra are measured with a PerkinElmer Lambda 365 spectrophotometer. PL spectra are acquired with a Princeton Instruments (HRS-300) monochromator integrated with a PIXIS CCD camera. TRPL profiles are acquired with a PicoQuant MPD single photon avalanche diode and a PicoHarp-300 timer. TRPL is probed at the PL peaks for each sample, and the wavelengths of probed PL are selected by the above HRS-300 monochromator. Picosecond pulse lasers are generated by a SC-PRO supercontinuum source and an AOTF system from Wuhan Yangtze Soton Laser Co., Ltd. PLQY measurements were carried out in an integrating sphere (10 cm diameter, Labsphere) following the approach described by de Mello et al. [64]. A 400 μm multimode fiber (Ocean Optics) was used to collect light from the sphere to couple into a spectrometer (Acton Spectrapro 500i), and a dispersion grating of 600 lp mm was used to disperse the light collected by the fiber onto an EMCCD camera (Andor Newton). The grating dispersion center wavelength is scanned across the wavelength range and the spectrum taken from each grating position is stitched together to give a broad wavelength range of 500 to 1200 nm.

### Density functional theory (DFT) calculations
DFT calculations are conducted with a Gaussian-16 software package. Molecular geometry is optimized at B3LYP/6-31 G(d,p) level. Time-dependent DFT is performed at ωB97X-D/6-31 G(d,p) level with a PCM of CF as solvent, and 30 states are calculated for each molecule[65].

### Transient absorption spectroscopy (TAS)
TAS is performed with an Ultrafast Systems Helios femtosecond transient absorption spectrometer. A femtosecond laser amplifier (Light Conversion) was used to generate a train of 1030 nm pulses, which were split into two beams to generate the pump and probe pulses, respectively. For probe, the pulses were focused onto a sapphire crystal and a YAG crystal to generate the visible (500–910 nm) and infrared (1100–1600 nm) continuum, respectively. An optical parametric amplifier was used to generate the pump beam centred at 750 nm (fluence ~3–5 μJ cm$^{-2}$). A mechanical delay stage was used to control time delay between pump and probe pulses.

### Molecular dynamics (MD) simulations
MD simulations are performed with a Gromacs-2019.3 software package. The simulations were carried out with the periodic boundary condition using the leap-frog integrator with a time step of 1.0 fs. A spherical cut-off of 1.2 nm for the summation of Van der Waals interactions and short-range Coulomb interactions and the particle-mesh Ewald method for solving long-range Coulomb interactions were used throughout. The velocity rescaling or Nose-Hoover thermostat was applied to control temperature and the Berendsen or Parrinello-Rahman barostat was used to control pressure. More details can be found in Supplementary Note 1.

### Grazing incidence wide angle X-ray scattering (GIWAXS)
GIWAXS measurements were performed at beamline 7.3.3, Advanced Light Source, Lawrence Berkeley National Laboratory[66]. The samples were measured in a helium environment to minimize air scattering using 10 keV energy X-rays, which was incident at a grazing angle of 0.12°. The scattered X-rays were detected using a Pilatus 2 M photon counting detector. The sample to detector distance was calibrated from diffraction peaks of the Silver-Behenate.

### Resonant soft X-ray scattering (R-SoXS)
R-SoXS measurements are performed at beamline 11.0.1.2, Advanced Light Source, Lawrence Berkeley National Laboratory[67]. The sample to detector distance is calibrated from diffraction peaks of polystyrene nanoparticles and beamline energy is calibrated by a fullerene-based sample. The beam size at the sample is ~100 μm × 200 μm, and two-dimensional R-SoXS patterns are collected on an in-vacuum CCD camera (Princeton Instrument PI-MTE) at −45 °C.

### Device fabrication and characterization
Devices are fabricated in conventional structure with ITO/PEDOT:PSS/active-layer/PNDIT-F3N/Ag. The effective device area is 0.04 cm$^2$, which is controlled by an iron aperture. The hole-transporting layer PEDOT:PSS (Al 4083) was spin-cast onto the ITO glass at 4000 rpm, and then thermally treated at 150 C° for 15 min. The solutions for active layers were prepared in chloroform with a total concentration of 16 mg mL$^{-1}$ and were stirred at 50 °C for ~30 min to fully dissolve. The D:A weight ratio was kept at 1:1.2. The solution was spin-cast on the PEDOT:PSS modified substrate at 3000 rpm. Three all-polymer systems were thermally annealed at 100 °C for 5 min. Then the PNDIT-F3N solution (1 mg mL$^{-1}$ in methanol) was spin-cast onto the active layer as electron transporting layer at 3000 rpm. At last, 160 nm Ag was thermally evaporated to the samples with a rate of 1–2 Å s$^{-1}$, at ~2 × 10$^{-6}$ Pa. The J–V characteristics were measured on a computer controlled Keithley 2400 source meter under illumination of an AM 1.5 G solar simulator (EnliTech SS-X), which was calibrated by a certified silicon reference cell. The photostability was evaluated with a customized solar cell lifetime testing system (Guangzhou Crysco Equipment Limited). The J–V characteristics were continuously measured under a LED light source matched with the AM 1.5 G

illumination. The testing is under *MPP* mode, and all samples were tested in the nitrogen gas glovebox.

## Reporting summary
Further information on research design is available in the Nature Portfolio Reporting Summary linked to this article.

## Data availability
The data that support the findings of this study are available in the HKU data repository (https://doi.org/10.25442/hku.24967422). Source data are provided with this paper.

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

## Acknowledgements

P.C.Y.C. acknowledges support from the Hong Kong Research Grant Council (27200822, 16300023), National Natural Science Foundation of China (22222905), and the University Research Council of the University of Hong Kong (HKU). Z.W. acknowledges support from Hong Kong Innovation and Technology Fund (MHP/064/20). Y.Y. acknowledges the support from the National Natural Science Foundation of China (22173108), and G.H. acknowledges the support from the Youth Innovation Promotion Association, CAS (2023037). H.-L.Y. acknowledges support from the Hong Kong Research Grant Council (11307323). X-ray data was acquired at the Advanced Light Source, Lawrence Berkeley National Laboratory, which was supported by the Director, Office of Science, Office of Basic Energy Sciences, of the US Department of Energy under contract no. DE-AC02-05CH11231. The authors acknowledge Dr Christopher C.S. Chan for his help with optical experiments. Publication made possible in part by support from the HKU Libraries Open Access Author Fund sponsored by the HKU Libraries.

## Author contributions

Z.W. performed the optical experiments, DFT calculations, device fabrication and data analysis of this project. Y.G. and X.L. assisted in sample preparation and optical experiments. W.S., G.H. and Y.Y. performed the MD simulations and analysis. K.D., S.M. and H.A. performed the X-ray scattering experiments. N.Z. and H.-L.Y. assisted with the device fabrication and photostability test. P.C.Y.C. conceived and supervised the project. P.C.Y.C. and Z.W. wrote the manuscript. All authors reviewed the manuscript.

## Competing interests

## Additional information

**Supplementary information** The online version contains Supplementary Material available at https://doi.org/10.1038/s41467-024-45455-0.

