## [Peer Review File · Nature Communications]

The role of interfacial donor–acceptor percolation in efficient and stable all-polymer solar cellsReviewer #1 (Remarks to the Author):

In this manuscript, Wang and co-workers investigate the charge generation mechanism and structure–property relationship of three high-performance all-polymer OPV blends based on polymerized-Y6 acceptors (Y6-PAs), and draw comparison to those found in blends based on the model small-molecule acceptor Y6. The authors used transient absorption spectroscopy to show that free charge generation happens at ~ 100 ps, and therefore a long (nanosecond) exciton lifetime in the low-gap acceptors is needed to facilitate efficient charge generation. While insufficient exciton lifetime is found in Y6-PA films, much extended exciton lifetime is found when the polymers are dispersed, indicating that dispersed Y6-PA chains are located at the percolated donor-acceptor interface. This picture is well supported by molecular dynamics (MD) simulations as well as resonant soft X-ray scattering (R-SoXS) experiments. Besides providing insights on the charge generation dynamics in Y6-PA blends, the results show that the all-polymer blends, thanks to the higher donor–acceptor miscibility, are much closer to the percolation threshold than those based on Y6 molecular acceptors. This in turn leads to a greater thermodynamic stability of the blend, which is known to be a major challenge for OPV based on Y6-type small-molecule acceptors.

Overall, this is a timely and solid study which will surely help the community to better understand the working mechanism of all-polymer solar cells, and inspire researchers especially those who work on material synthesis and device engineering to design more efficient and stable organic solar cells. The scope and novelty of this manuscript is suitable to Nature Communications, and the conclusions are well supported by the experimental and simulation results. I would be happy to recommend this manuscript be accepted for publication after some minor revisions and discussions. My detailed comments and suggestions are as follows:

1. While the authors found that charge generation takes ~ 100 ps under 750 nm excitation, which selectively excites the acceptors, it is not clear if similar charge separation dynamics are found under other excitation wavelengths (e.g., when the donors are selectively excited). This is important since both donor and acceptor are excited under device operation. The authors should provide additional transient absorption data under other excitation wavelengths to cover this point.
2. The composition variation (domain purity) information was extracted from R-SoXS data acquired at 284 eV. While it seems that R-SoXS profiles acquired at 284.7 eV show higher intensities/contrasts (Figure S25-26). The authors should clarify this and compare the composition information extracted from data acquired at different energies.
3. According to previous report by Brabec et al. (Nat. Energy 2020, 5, 711–719), a long exciton lifetime is very important to allow efficient charge generation at the D–A interfaces in systems with small or negligible donor–acceptor energy offsets. However, the energy level offsets of the studied PM6:Y6-PA systems are not provided in the manuscript. The authors should include additional discussions in the manuscript to address this point.
4. It is noted that some recent studies indicated that the choice of donor polymers can also play a key role on the blend morphology stability, e.g. Joule 2023, 7, 810-829. The authors should add some discussions about this point in the manuscript.

Reviewer #2 (Remarks to the Author):

Zhen Wang et al. present a study of exciton dissociation dynamics and morphology in all-polymer organic solar cells. As acceptor materials, they used polymer derivatives of the small molecule acceptor Y6, that have been shown to combine high efficiency with enhanced stability against morphological degradation in previously published works. These polymer acceptors are blended with the PM6 donor polymer; comparison with the "standard" PM6:Y6 blend is also given in all cases.

From femtosecond transient absorption studies, the authors find that hole transfer from the acceptor to the donor occurs on the tens of picoseconds time scale, justifying the need for nanosecond exciton lifetimes to support near unity exciton dissociation yields. From transient photoluminescence, they find that polymer acceptors in their solid state show strong aggregation-induced quenching and therefore can't sustain nanosecond exciton lifetimes. However, due to good miscibility with PM6, the polymer acceptors avoid aggregation at the donor-acceptor interface, thus avoiding aggregation-induced quenching. The good miscibility of the polymer acceptors with PM6 also reduces the trend for donor:acceptor demixing, causing electrical performance degradation in organic solar cells comprising PM6:Y6 blends.

There is no doubt that this study is timely and the objective - enacting control over morphology to match efficiency and stability in organic photovoltaics - is of high importance to the field and to society. However, in my opinion, both the generality and the validity of the key claims have not been sufficiently supported by evidence to demonstrate new design principles. For this reason, I do not think the manuscript is publishable in Nature Communications in its present form. On one hand, the results might be valid only for blends with PM6; on the other hand they might be valid beyond all-polymer solar cells and refer also to small molecule acceptors. I suggest a clarification of the scope and validity of the key claims before this paper can be published in Nature Communications.

Detailed analysis:

[1] Figure 1e: the HOMO and LUMO wavefunctions of PY-IT show very little spatial overlap; therefore the contribution of this configuration to the transition moment of the S₀-S₁ transition should be rather low. For a fair comparison with the small molecule Y6, I suggest to represent the configuration with the highest contribution to the transition moment of the S₀-S₁ transition.

[2] Page 9, line 169: The authors find strong aggregation-induced quenching (increase of k_{nr}) in the polymer Y6 while small molecule Y6 does not show this effect. Although all three polymer Y6 show this effect, the molecular structures show that all polymer Y6 contain tert-butyl groups. The authors should comment whether the notion of aggregation-induced quenching is intrinsic to polymer acceptors, or whether the phenomenon would need to be handled also in side chain engineering for small molecule acceptors.

[3] Page 9, line 181: "near acceptor films": do the authors mean "neat acceptor films"?

[4] page 10, line 191. The authors show average numbers for contacting D-A pairs of 2.68 and 3.27 for PM6:Y6 and PM6:PY-IT, respectively, after accounting for the difference in molecular weights. However, in the supporting information, it has not become clear how this normalization to molecular weights has been achieved. Could the authors clarify this?

[5] Does a difference of less than 20% between the contacting numbers of PM6:Y6 and PM6:PY-IT justify the notion of an entirely different morphology (aggregated versus intercalated, respectively)?

[6] Figure 4c: according to this figure, the polymer acceptor shows better mixing with PM6 than small molecule Y6. Can this notion be generalized to other donor polymers than PM6? Is the better mixing due to the acceptor being a polymer, or due to the tert-butyl groups present in the polymer acceptors?

[7] Figure 4d: in this figure panel, the authors depict a "Y6-PA" phase on green background. However, according to Figure 2c, such a pure Y6-PA phase should present fast exciton quenching; excitons created in the pure Y6-PA phase would have low probability to diffuse to the interface before being quenched. However, in the absence of such a pure Y6-PA phase, typically one would expect fast exciton breaking but also fast geminate recombination. Indeed, Fig. 2b seems to indicate fast polaron loss after 100 ps. But that would contradict the notion of high efficiency, indicated in the title of the manuscript. Could the authors resolve these apparent contradictions?

[8] From Figure 4c, the author infer a greater stability of polymer acceptors against morphological

degradation. However, morphological degradation does not only involve the donor-acceptor blend but also the extracting layers. Different interactions can lead to vertical demixing or changes of local alignment. Could the authors comment on the role of polymer acceptors with respect to these aspects of morphological degradation?

Reviewer #3 (Remarks to the Author):

Please refer to the attached file

Reviewer #3 Attachment on the following page

The study conducted by Chow et al. described an intriguing investigation into the Y6-type small molecule acceptors (Y6-SMAs) and all-polymer blends based on polymerized Y6-type acceptors (Y6-PAs) in underlying their charge generation dynamics and structure–property relationships. The authors suggest that the well dispersed Y6-PA showed a much shorter longer exciton lifetimes (~1.1-1.9 ns) than aggregated polymers in films (~0.3-0.5 ns), taking the advantages of the aggregated polymers Y6-PAs have larger miscibility with the donor polymer (PM6) than Y6-SMAs, leading to D–A percolation that effectively prevents the formation of Y6-PA aggregates at the interface. Therefore, the interfacial D–A percolation plays a key role in suppressing interfacial charge recombination to enable efficient charge generation.

Overall, this manuscript is well-organized and well-written. However, some of the comments overclaim with insufficient experimental support. Major revisions and supplementary data are needed before we would consider recommending this manuscript for publication in *Nature Communications*. Specific comments are as follows.

Major issues:

1. Please provide the full characterization data of the polymerized acceptors, including PY-IT, PYF-T-o, and PY-V-CE \geq , to show the identities of the polymers.
2. What is the reason for using chloroform as the solvent in the emission study? Additionally, the authors mentioned that the exciton lifetime of the polymers is strongly dependent on the degree of aggregation. Please provide evidence of the degree of aggregation in solution by measuring their VT UV-vis and emission spectra.
3. XRD and GIWAXS measurements should be included to provide the packing efficiency of Y6-SMAs and Y6-PAs, in addition to the MD simulation.
4. The authors mentioned that when these Y6-PAs are blended with PM6, a blue shift in emission was observed. However, the blue shift of these Y6-PAs is so small. Please further elaborate on their differences in terms of molecular packing, providing supportive evidence.
5. As observed from the emission spectra of Y6, as shown in Figure S16,

It is obvious that the emissive state has changed with increasing doping concentration, as evidenced by the red shift in the emission maximum of Y6 from 800 nm to 925 nm. Additionally, the emission profile and emission energy of Y6-SMAs and Y6-PAs are completely different, as shown in Figure S17, indicating that the emission of these compounds originates from different excited states. Therefore, comparing the emission lifetimes of these two distinctive emissive states is not an appropriate approach, as they belong to two completely different classes of molecules. Basically, they are totally two different classes of molecules without any point to compare.

Figure S17. PL spectra comparison between pristine acceptor and blended films, excited at 720 nm.

6. Please fabricate the device show that the PCE and operation stability are indeed improved after all these characterisation and MD calculation.

Reviewer #1:

In this manuscript, Wang and co-workers investigate the charge generation mechanism and structure–property relationship of three high-performance all-polymer OPV blends based on polymerized-Y6 acceptors (Y6-PAs), and draw comparison to those found in blends based on the model small-molecule acceptor Y6. The authors used transient absorption spectroscopy to show that free charge generation happens at ~100 ps, and therefore a long (nanosecond) exciton lifetime in the low-gap acceptors is needed to facilitate efficient charge generation. While insufficient exciton lifetime is found in Y6-PA films, much extended exciton lifetime is found when the polymers are dispersed, indicating that dispersed Y6-PA chains are located at the percolated donor-acceptor interface. This picture is well supported by molecular dynamics (MD) simulations as well as resonant soft X-ray scattering (R-SoXS) experiments. Besides providing insights on the charge generation dynamics in Y6-PA blends, the results show that the all-polymer blends, thanks to the higher donor–acceptor miscibility, are much closer to the percolation threshold than those based on Y6 molecular acceptors. This in turn leads to a greater thermodynamic stability of the blend, which is known to be a major challenge for OPV based on Y6-type small-molecule acceptors.

Overall, this is a timely and solid study which will surely help the community to better understand the working mechanism of all-polymer solar cells, and inspire researchers especially those who work on material synthesis and device engineering to design more efficient and stable organic solar cells. The scope and novelty of this manuscript is suitable to *Nature Communications*, and the conclusions are well supported by the experimental and simulation results. I would be happy to recommend this manuscript be accepted for publication after some minor revisions and discussions. My detailed comments and suggestions are as follows:

Author response:

Many thanks for the reviewer's positive comments.

1. While the authors found that charge generation takes ~100 ps under 750 nm excitation, which selectively excites the acceptors, it is not clear if similar charge separation dynamics are found under other excitation wavelengths (e.g., when the donors are selectively excited). This is important since both donor and acceptor are excited under device operation. The authors should provide additional transient absorption data under other excitation wavelengths to cover this point.

Author response:

Thanks for pointing this out. Indeed, both donor and acceptor are excited under PV device operation. We have performed additional transient absorption (TA) data for the blended films under 550 nm excitation, which excites both donor and acceptor. It is found that there is little difference in the overall charge generation kinetics compared to the 750 nm excitation (both take ~100 ps to complete). This can be seen from the TA spectra (visible region) of the four blended films excited at 550 nm shown in **Figure R1** below, and the decay kinetics of the ground state bleaching (GSB) of PM6 are summarized in **Figure R2**. Though the relative intensities are different in these blended films, the decay kinetics are similar, with the growth in polaron photoinduced absorption (charge generation) taking ~100 ps to complete for all four blends. We have now added Figure R1-R2 to the revised Supplementary Information as new Supplementary Figure S11-S12 and added the relevant discussion to the revised manuscript.

Figure R1. TA spectra (visible region) of PM6:Y6 (a), PM6:PY-IT (b), PM6:PYF-T-o (c), and PM6:PY-V- γ (d) blends, excited at 550 nm.

Figure R2. Integrated normalized TA kinetics at 620-650 nm for PM6 blended films, excited at 550 nm.

2. The composition variation (domain purity) information was extracted from R-SoXS data acquired at 284 eV. While it seems that R-SoXS profiles acquired at 284.7 eV show higher intensities/contrasts (Figure S25-26). The authors should clarify this and compare the composition information extracted from data acquired at different energies.

Author response:

Thanks for the suggestion. The X-ray energy (284 eV) used for analysis in the manuscript is closer to the carbon K-edge (~ 283.8 eV). Therefore, R-SoXS profiles acquired at this energy have better material contrasts and lower fluorescence signals than those at higher energies (*J. Mater. Chem. C* **2013**, *1*, 187-201; *Phys. Rev. Lett.* **2017**, *119*, 167801).

In order to check the energy-dependence of R-SoXS data, we further analysed R-SoXS profiles acquired at 284.7 eV and summarized in **Figure R3** below. By normalizing to the data for PM6:Y6 film (taking it as 1), it is estimated that the root-mean-square composition variations for PM6:PY-IT, PM6:PYF-T-o and PM6:PY-V- γ films are 0.77, 0.77 and 0.82, respectively. This is fairly similar to the data we got at 284 eV, which again indicates that PM6:Y6 blend has better domain purity than the other all-polymer blends.

Figure R3. Lorentz-corrected thickness-normalized R-SoXS profiles of neat (a) and blended (b) films, acquired at 284.7 eV.

3. According to previous report by Brabec et al. (*Nat. Energy* **2020**, *5*, 711–719), a long exciton lifetime is very important to allow efficient charge generation at the D–A interfaces in systems with small or negligible donor–acceptor energy offsets. However, the energy level offsets of the studied PM6:Y6-PA systems are not provided in the manuscript. The authors should include additional discussions in the manuscript to address this point.

Author response:

Thanks for pointing this out. The energy level (HOMO) offset estimated from cyclic voltammetry method for the three all-polymer blends are 0.16, 0.15 and 0.12 eV respectively (*Adv. Mater.* **2022**, *34*, 2108749; *Adv. Sci.* **2022**, *9*, 2202513; *Adv. Mater.* **2022**, *34*, 2200361). The HOMO offset of these all-polymer blends falls into the charge-transfer-state–exciton (CT–LE) equilibrium regime that requires long exciton lifetime for decent external quantum efficiency (EQE), as described and discussed in the study mentioned by the reviewer (*Nat. Energy* **2020**, *5*, 711–719). The relationships of non-radiative energy loss, EQE and HOMO offset are shown in **Figure R4**. Moreover, though HOMO offset of these all-polymer blends is slightly larger than that of PM6:Y6 blend (0.09 eV, *Joule* **2019**, *3*, 1140-1151), our TA data indicates that it takes the same timescale (~ 100 ps) for charge generation to complete for PM6:Y6 and the three all-polymer blends. This implies that the slightly larger HOMO offset doesn't help much with the charge generation process, and hence a long exciton lifetime is

necessary for these all-polymer blends as well. We have added relevant discussion in our revised manuscript to address this point.

Figure R4. Non-radiative energy loss ($\Delta V_{OC,non-rad}$) and maximum EQE of solar cells studied with respect to HOMO offset (ΔE_{HOMO}). Borrowed from ref. [*Nat. Energy* **2020**, 5, 711–719].

4. It is noted that some recent studies indicated that the choice of donor polymers can also play a key role on the blend morphology stability, e.g., *Joule* **2023**, 7, 810-829. The authors should add some discussions about this point in the manuscript.

Author response:

Thanks for the comment, we agree that the choice of donor polymer is also an important factor for the overall morphology stability. We have added the relevant discussion in our revised manuscript, and in Supplementary Note 2 of the revised Supplementary Information for extended discussions.

Added discussion in main text:

“Nevertheless, it should be noted that besides blend morphology stability, other factors such as polymer donor degradation and interfacial defects are also likely to affect the overall device stability. Therefore, in addition to developing strategies to improve morphological stability, future research should also target to overcome these factors in order to achieve highly stable and efficient OPV devices that can meet industry requirements.”

Reviewer #2:

Zhen Wang et al. present a study of exciton dissociation dynamics and morphology in all-polymer organic solar cells. As acceptor materials, they used polymer derivatives of the small molecule acceptor Y6, that have been shown to combine high efficiency with enhanced stability against morphological degradation in previously published works. These polymer acceptors are blended with the PM6 donor polymer; comparison with the "standard" PM6:Y6 blend is also given in all cases.

From femtosecond transient absorption studies, the authors find that hole transfer from the acceptor to the donor occurs on the tens of picoseconds time scale, justifying the need for nanosecond exciton lifetimes to support near unity exciton dissociation yields. From transient photoluminescence, they find that polymer acceptors in their solid state show strong aggregation-induced quenching and therefore can't sustain nanosecond exciton lifetimes. However, due to good miscibility with PM6, the polymer acceptors avoid aggregation at the donor-acceptor interface, thus avoiding aggregation-induced quenching. The good miscibility of the polymer acceptors with PM6 also reduces the trend for donor:acceptor demixing, causing electrical performance degradation in organic solar cells comprising PM6:Y6 blends.

There is no doubt that this study is timely and the objective - enacting control over morphology to match efficiency and stability in organic photovoltaics - is of high importance to the field and to society. However, in my opinion, both the generality and the validity of the key claims have not been sufficiently supported by evidence to demonstrate new design principles. For this reason, I do not think the manuscript is publishable in *Nature Communications* in its present form. On one hand, the results might be valid only for blends with PM6; on the other hand, they might be valid beyond all-polymer solar cells and refer also to small molecule acceptors. I suggest a clarification of the scope and validity of the key claims before this paper can be published in *Nature Communications*.

Author response:

Many thanks for the reviewer's overall positive comments. The suggestions from the reviewer surely help a lot to improve our work. In the revision manuscript, we have included additional experimental data and discussions to address the reviewer's comments.

Detailed analysis:

[1] Figure 1e: the HOMO and LUMO wavefunctions of PY-IT show very little spatial overlap; therefore, the contribution of this configuration to the transition moment of the S_0 - S_1 transition should be rather low. For a fair comparison with the small molecule Y6, I suggest to represent the configuration with the highest contribution to the transition moment of the S_0 - S_1 transition.

Author response:

Thanks for pointing this out. By further analysing our DFT results, it should be noted that while the S_0 - S_1 transition for Y6 is indeed coming from HOMO to LUMO orbital transition, for Y6-PAs, there are multiple orbital transitions that contribute to S_0 - S_1 transition. For PY-IT and PYF-T-o, we find that HOMO-1 to LUMO contributes mainly to the S_0 - S_1 transition, and for PY-V- γ , HOMO to LUMO contributes mainly to the S_0 - S_1 transition along with others. We have summarized the orbital transitions that contribute to S_0 - S_1 transition of the four acceptors in **Table R1** below. Also, we have obtained the HOMO-1 orbitals and added them in Figure 1 (with updated figure caption and main text), as well shown in **Figure R5** below.

Table R1. Orbital transitions that contribute to $S_0 \rightarrow S_1$ transition of four acceptors.

Y6 ($S_0 \rightarrow S_1$)	PY-IT ($S_0 \rightarrow S_1$)	PYF-T-o ($S_0 \rightarrow S_1$)	PY-V- γ ($S_0 \rightarrow S_1$)
HOMO -- LUMO (*)	HOMO-3 -- LUMO	HOMO-3 -- LUMO	LUMO-3 -- LUMO+1
	HOMO-3 -- LUMO+2	HOMO-3 -- LUMO+1	HOMO-3 -- LUMO+2
	HOMO-3 -- LUMO+3	HOMO-3 -- LUMO+3	HOMO-2 -- LUMO
	HOMO-2 -- LUMO+1	HOMO-2 -- LUMO+1	HOMO-2 -- LUMO+2
	HOMO-2 -- LUMO+3	HOMO-2 -- LUMO+2	HOMO-1 -- LUMO+1
	HOMO-1 -- LUMO (*)	HOMO-1 -- LUMO (*)	HOMO -- LUMO (*)
	HOMO-1 -- LUMO+2	HOMO-1 -- LUMO+2	
	HOMO -- LUMO	HOMO -- LUMO	
	HOMO -- LUMO+1	HOMO -- LUMO+1	

(*) Main contribution.

**Figure R5.** Frontier orbitals (HOMO, LUMO and HOMO-1) of four acceptors.

[2] Page 9, line 169: The authors find strong aggregation-induced quenching (increase of k_{nr}) in the polymer Y6 while small molecule Y6 does not show this effect. Although all three polymer Y6 show this effect, the molecular structures show that all polymer Y6 contain tert-butyl groups. The authors should comment whether the notion of aggregation-induced quenching is intrinsic to polymer acceptors, or whether the phenomenon would need to be handled also in side chain engineering for small molecule acceptors.

Author response:

Thanks for the insightful comment. To reveal whether the aggregation-induced quenching is intrinsic to these polymer acceptors, as opposed to the choice of side chains, we have introduced an additional material system in our study: PY-monomer. As shown in the chemical structure (**Figure R6**), PY-monomer has the same side chains as the Y6-PAs studied in this work. **Figure R7** shows the PL and TRPL data for the PY-monomer, and **Figure R8** and **R9** show the TA data for PY-monomer neat and blended films. It is clear that the PY-monomer behaves very similarly to Y6, even though they have different side chains. This provides evidence that the aggregation-caused quenching comes from the polymerization of these systems. We have added the relevant data (as new Supplementary Figure S26-29) and discussions in the revised main text and Supplementary Information.

Added discussion in main text:

“We also performed TA and TRPL for the PY-monomer, which has the same side chains as Y6-PAs studied in this work (see Supplementary Fig. S26-29). We observe little difference in exciton dynamics between the PY-monomer and Y6 films, which implies that the changes in exciton dynamics observed in the Y6-PAs are results of polymerization rather than the side chains.”

Figure R6. Molecular structure of PY-monomer.

Figure R7. (a) Normalized PL spectra for PY-monomer solution/film samples and PM6:PY-monomer blend film. (b) TRPL data for PY-monomer solution and film samples.

Figure R8. TA data for neat PY-monomer film (a-b), and PM6:PY-monomer blended film (c-d).

Figure R9. Integrated decay kinetics for GSB features of PY-monomer (a) and charge generation (hole transfer) features of PM6:PY-monomer blend (b).

[3] Page 9, line 181: "near acceptor films": do the authors mean "neat acceptor films"?

Author response:

Thanks for pointing this out. Yes, we meant “neat acceptor films”, and the typo is now corrected in the revision manuscript.

[4] Page 10, line 191. The authors show average numbers for contacting D–A pairs of 2.68 and 3.27 for PM6:Y6 and PM6:PY-IT, respectively, after accounting for the difference in molecular

weights. However, in the supporting information, it has not become clear how this normalization to molecular weights has been achieved. Could the authors clarify this?

Author response:

Thanks for the comment. The simulated blending systems of PM6:Y6 and PM6:PY-IT contain 444 Y6 molecules or 60 PY-IT chains (each with 6 repeat units) and 44 PM6 chains (each with 10 repeat units), consistent with the D:A weight ratio of 1:1.2 in experiments. The total count of contacting D–A pairs (with contacting atoms more than 6 for each pair) for PM6:Y6 and PM6:PY-IT are 1191 and 1178, respectively. However, PM6:PY-IT has less acceptor units than PM6:Y6 due to the polymerization. Therefore, for a fair comparison, the average number of contacting D–A pairs was calculated by dividing the total count of contacting D–A pairs by the number of acceptor units. This also represents the average neighbours of donor units for each acceptor unit. We have added this clarification in Supplementary Note 1.

[5] Does a difference of less than 20% between the contacting numbers of PM6:Y6 and PM6:PY-IT justify the notion of an entirely different morphology (aggregated versus intercalated, respectively)?

Author response:

Thanks for the comment. Based on the good agreement between our MD simulation results and experimental R-SoXS results, we believe that the difference of ~18-20% between the contact atom numbers from MD simulation can indeed directly reflect the large difference in the overall blend morphology.

[6] Figure 4c: according to this figure, the polymer acceptor shows better mixing with PM6 than small molecule Y6. Can this notion be generalized to other donor polymers than PM6? Is the better mixing due to the acceptor being a polymer, or due to the tert-butyl groups present in the polymer acceptors?

Author response:

Thanks for the comment. We have included another donor polymer to our study (D18-Cl, molecular structure shown in **Figure R10**). We select D18-Cl because it can also be blended with the acceptor materials of this study to achieve high OPV efficiencies (both Y6-SMAs and Y6-PAs). We have carried out TA (**Figure R11-R12**) and R-SoXS characterizations (**Figure R13**) for the D18-Cl systems. Though the R-SoXS dataset was acquired at a different facility (Brookhaven National Laboratory) and the q range is slightly different from the PM6 systems, we can still compare the domain purities of these D18-Cl blends with their root-mean-square composition variations, where for D18-Cl:Y6, D18-Cl:PY-IT, D18-Cl-PYF-T-o, and D18-Cl:PY-V- γ are 1, 0.84, 0.86 and 0.86, respectively. This confirms our conclusion that a greater D-A miscibility is general to these high-performance all-polymer systems based on Y6-PAs. We have included the additional data and discussions of the D18-Cl systems in the revised main text and Supplementary Information (Supplementary Figure S42-45).

Added discussion in main text:

“We also perform optical and morphology characterizations for another donor polymer (D18-Cl) blended with Y6 and the Y6-PAs (see Supplementary Fig. S42-45). Note that high device efficiencies are achieved in these D18-Cl-based blends (~17-18%).^{53,54} Similar to the PM6 blends, in R-SoXS results the root-mean-square composition variations for D18-Cl:PY-IT, D18-

Cl-PYF-T-o, and *D18-Cl:PY-V-γ* are 0.84, 0.86 and 0.86 respectively after normalization to the data of *D18-Cl:Y6*.”

Figure R10. Molecular structure of D18-Cl.

Figure R11. TA spectra (visible region) of D18-Cl:Y6 (a), D18-Cl:PY-IT (b), D18-Cl:PYF-T-o (c), and D18-Cl:PY-V- γ (d) blends, excited at 750 nm.

Figure R12. Integrated normalized TA kinetics at 590-600 nm for D18-Cl blended films, excited at 750 nm. Similar to PM6 blends, charge generation in these D18-Cl blends takes ~ 100 ps to complete as well. This implies that not only in PM6, but a long enough exciton lifetime is also needed for the NFAs in other polymer donor blends.

Figure R13. Lorentz-corrected thickness-normalized R-SoXS profiles of D18-Cl:acceptor blended films, acquired at 284 eV.

[7] Figure 4d: in this figure panel, the authors depict a "Y6-PA" phase on green background. However, according to Figure 2c, such a pure Y6-PA phase should present fast exciton quenching; excitons created in the pure Y6-PA phase would have low probability to diffuse to the interface before being quenched. However, in the absence of such a pure Y6-PA phase, typically one would expect fast exciton breaking but also fast geminate recombination. Indeed, Fig. 2b seems to indicate fast polaron loss after 100 ps. But that would contradict the notion of high efficiency, indicated in the title of the manuscript. Could the authors resolve these apparent contradictions?

Author response:

Thanks for the insightful comment. First, we clarify that Figure 4d (now 4c) in the manuscript is just a schematic illustration of the (nanoscale) D–A mixing at the interface, and does not mean that the Y6-PA phase is excessively large and pure (in fact the opposite is concluded based on the presented R-SoXS and MD results). Based on the high efficiency (PCE >15% and EQE >75%), we know that vast majority of excitons can reach the D–A interface and separate. Indeed, we do see faster polaron loss after 100 ps in TA results, and we believe this is in line with lower FF of the Y6-PA blends. This is likely due to reduced domain purity, as we

uncovered in R-SoXS results, slightly increasing recombination losses during charge transport. We believe that future research should target to address this increase in recombination losses caused by the reduced domain purity.

Added discussion in main text:

“The reduced domain purity that we revealed for the Y6-PA blends is consistent with their faster carrier decay (Fig. 2b) and lower device PCE (~13.2–15.5%) compared to Y6 blend (~16.0%). Nevertheless, these Y6-PA blends are still among the most efficient all-polymer OPV systems to date, and we consider that high intra-chain electron transport in these materials helps to reduce recombination losses by allowing charges to move away from the mixed D–A interface. Engineering of the intra-chain charge transport for both polymer donor and acceptor materials may provide a design pathway to further suppress recombination losses in all-polymer blends.”

[8] From Figure 4c, the authors infer a greater stability of polymer acceptors against morphological degradation. However, morphological degradation does not only involve the donor–acceptor blend but also the extracting layers. Different interactions can lead to vertical demixing or changes of local alignment. Could the authors comment on the role of polymer acceptors with respect to these aspects of morphological degradation?

Author response:

Thanks for the comment. We agree that other factors are also important to the device stability. To further investigate the structure–device relationship, also in response to Comment #6 by Reviewer #3, we have fabricated device samples and did controlled photostability testing with the same device architecture (which rules out the influence of the extracting layers), and it is clear the all-polymer systems show better photostability (less “burn-in”) than the PM6:Y6 system thanks to the more stable D–A interfacial morphology. We have added additional device data and discussions to our revised manuscript and Supplementary Information. An in-depth investigation of all these factors is, however, beyond the scope of this present work.

Added discussion in main text:

“For device stability, it is clearly shown that PM6:Y6 device suffered from a more rapid decay in efficiency upon solar illumination compared to the three PM6:Y6-PA blends. This rapid efficiency drop is attributed to the device “burn-in”, which is often observed in non-fullerene SMA-based blends with hypo-miscible D–A morphology (over-purification) at the interfaces. Thanks to the increased D–A miscibility and interfacial percolation in the Y6-PA blends, these all-polymer blends suffer from reduced “burn-in” losses and thus an improved device stability over their SMA counterpart. We consider that the increased D–A miscibility in Y6-PA-based blends helps to explain the overall improvement in device stability compared to Y6-SMA-based blends (see Supplementary Note 3 for extended discussion). Nevertheless, it should be noted that besides blend morphology stability, other factors such as polymer donor degradation and interfacial defects are also likely to affect the overall device stability. Therefore, in addition to developing strategies to improve morphological stability, future research should also target to overcome these factors in order to achieve highly stable and efficient OPV devices that can meet industry requirements.”

Reviewer #3:

The study conducted by Chow et al. described an intriguing investigation into the Y6-type small molecule acceptors (Y6-SMAs) and all-polymer blends based on polymerized Y6-type acceptors (Y6-PAs) in underlying their charge generation dynamics and structure–property relationships. The authors suggest that the well dispersed Y6-PA showed a much shorter longer exciton lifetimes (~1.1-1.9 ns) than aggregated polymers in films (~0.3-0.5 ns), taking the advantages of the aggregated polymers Y6-PAs have larger miscibility with the donor polymer (PM6) than Y6-SMAs, leading to D–A percolation that effectively prevents the formation of Y6-PA aggregates at the interface. Therefore, the interfacial D–A percolation plays a key role in suppressing interfacial charge recombination to enable efficient charge generation.

Overall, this manuscript is well-organized and well-written. However, some of the comments overclaim with insufficient experimental support. Major revisions and supplementary data are needed before we would consider recommending this manuscript for publication in *Nature Communications*. Specific comments are as follows.

Author response:

Many thanks for the reviewer’s positive comments. We have carried out additional experiments and revised the manuscript according to the reviewers’ comments.

Major issues:

1. Please provide the full characterization data of the polymerized acceptors, including PY-IT, PYF-T-o, and PY-V- γ , to show the identities of the polymers.

Author response:

Thanks for the comment. The three Y6-PAs are commercially available, and we purchased them from companies (i.e., Hyper Chem Limited and eFlexPV Limited). The routine characterization of the three Y6-PAs including UV-vis absorption, PL, TRPL, TA are shown in Figures 1b-c, 2c, S13 and S7-S8, respectively in the manuscript and Supplementary Information. Also, the concentration-dependent UV-vis absorption, PL and TRPL results are added in Figures S15-S21 in the answer to Question #2. As for the molecular weights, we borrowed the gel permeation chromatography (GPC) characterization from the references that reported the Y6-PAs originally, and summarized them in **Figure R14** below. We note that, according to the vendors, the Y6-PAs were synthesized with the same method in the references and the molecular weights are basically the same as in the references. This is also supported by the similar device PCEs that we measured in our lab, as summarized in **Table R3** in the answer to Question #6.

Figure R14. Gel permeation chromatography (GPC) characterization of PY-IT (a), PYF-T-o (b), and PY-V- γ (c), borrowed from refs. [*Adv. Mater.* **2020**, *32*, 2005942; *Angew. Chem. Int. Ed.* **2021**, *60*, 10137; *Adv. Mater.* **2022**, *34*, 2200361.]

2. What is the reason for using chloroform as the solvent in the emission study? Additionally, the authors mentioned that the exciton lifetime of the polymers is strongly dependent on the degree of aggregation. Please provide evidence of the degree of aggregation in solution by measuring their VT UV-vis and emission spectra.

Author response:

Thanks for the comment. First, we would like to clarify that we believe there is little aggregations in solutions, and therefore comparing the data in solution and film allows us to study the effect of aggregation on the material's optical properties in this work. We used chloroform (CF) because it is also used for device fabrication of these systems. To extend our study, we also performed experiments in chlorobenzene (CB), which will allow us to study the effect of solvent polarity on the polymer chains. The concentration-dependent UV-vis absorption, PL spectra, and TRPL data for the four acceptors in CB solutions are shown in **Figures R15-17**, respectively. We find that the data are basically the same as CF solutions.

We also performed temperature-dependent UV-vis absorption spectra for CB solutions (0.005 mg/mL), see **Figure R18** below. At higher temperatures, the absorption spectra of solutions samples get broadened due to the stronger electron-phonon couplings and likely molecular relaxation. It is observed that there is a slight blueshift for Y6 solution as the temperature ramps up, while this shift is not observed on three Y6-PA solution samples. These results therefore indicate that with such low concentration, the degree of aggregation of Y6-PAs in solution is negligible. These solution results are added in the revised Supplementary Information as new Supplementary Figure S15-S21.

Figure R15. Normalized UV-vis absorption spectra of Y6, PY-IT, PYF-T-o and PY-V- γ in CB solutions.

Figure R16. Normalized PL spectra of Y6, PY-IT, PYF-T-o and PY-V- γ in CB solution.

Figure R17. Concentration-dependent TRPL profiles of Y6, PY-IT, PYF-T-o and PY-V- γ in CB solutions.

Figure R18. Temperature-dependent UV-vis absorption spectra of Y6, PY-IT, PYF-T-o and PY-V- γ in CB solutions (0.005 mg/mL).

3. XRD and GIWAXS measurements should be included to provide the packing efficiency of Y6-SMAs and Y6-PAs, in addition to the MD simulation.

Author response:

Thanks for the comment. We have performed additional GIWAXS measurements. Two-dimensional (2D) GIWAXS patterns of neat and blended films are displayed in **Figure R19** and **R20**, respectively below. And their one-dimensional (1D) cuts along in-plane and out-of-plane directions are summarized in **Figure R21**.

It is clear all the neat and blended films show preferred face-on orientation with respect to substrates, by recognizing the (010) peaks along out-of-plane direction. We have fitted the (010) peaks along out-of-plane direction of the 1D profiles and summarized the q value, π - π stacking distance (d), full-width at half-maximum (FWHM), coherence length (L_C), for neat and blended films in **Table R2**. For the neat films, the π - π stacking distance of Y6 is $\sim 3.55 \text{ \AA}$, and of Y6-PAs is $\sim 3.81 \text{ \AA}$. The coherence length of Y6 packing is $\sim 23.5 \text{ \AA}$, and that of Y6-PAs is ~ 14.9 - 17.1 \AA . As for the blended films with PM6, the π - π stacking distance and coherence length changed a bit, while that of Y6-SMA is still much larger than the Y6-PAs. This indicates that Y6-SMAs do have greater packing efficiencies than Y6-PAs, in both neat and blended films. We have included these new GIWAXS results in the revised manuscript and Supplementary Information (as Figures S37-S39 and Table S5).

Added discussion in main text:

“We first perform grazing-incidence wide-angle X-ray scattering (GIWAXS) characterization for both neat and blended films. The results are displayed in Supplementary Fig. S37-39 and Table S5. Overall, we find evidence that Y6-PAs are less ordered than Y6 in both neat and blended films.”

Figure R19. GIWAXS 2D patterns of neat films.

Figure R20. GIWAXS 2D patterns of blended films.

Figure R21. GIWAXS 1D profiles along in-plane and out-of-plane directions of neat (a) and blended (b) films.

Table R2. Summary of the GIWAXS results ((010) peaks along out-of-plane direction), including q value, π - π stacking distance (d), full-width at half-maximum (FWHM), coherence length (L_c), for neat and blended films.

(010) peaks of neat films				
Sample	Y6	PY-IT	PYF-T-o	PY-V- γ
q (\AA^{-1})	1.77	1.65	1.65	1.65
d (\AA)	3.55	3.81	3.81	3.81
FWHM (\AA^{-1})	0.24	0.33	0.36	0.38
L_c (\AA)	23.5	17.1	15.7	14.9
(010) peaks of blended films				
Sample	PM6:Y6	PM6:PY-IT	PM6:PYF-T-o	PM6:PY-V- γ
q (\AA^{-1})	1.75	1.68	1.68	1.68
d (\AA)	3.59	3.74	3.74	3.74
FWHM (\AA^{-1})	0.28	0.40	0.42	0.41
L_c (\AA)	20.2	14.1	13.4	13.8

4. The authors mentioned that when these Y6-PAs are blended with PM6, a blue shift in emission was observed. However, the blue shift of these Y6-PAs is so small. Please further elaborate on their differences in terms of molecular packing, providing supportive evidence.

Author response:

Thanks for the comment. As we discuss in the manuscript, the small red-shifting from solution (dispersed) to film (aggregated) for the Y6-PAs is because of their intrachain exciton delocalization. In contrast, Y6 molecules show much greater red-shift between solution and films because of the strong intermolecular interactions in these systems (see new Supplementary Figure S15). The small red-shift for Y6-PAs is one evidence for weak intermolecular interactions in these polymers. When the Y6-PAs are blended with PM6, the emission is slightly blue shifted, which indicates that they are slightly more dispersed in the PM6 blend (i.e., at the D–A interface). This is indeed confirmed by our detailed MD and R-SoXS studies, as described in the manuscript.

5. As observed from the emission spectra of Y6, as shown in Figure S16, it is obvious that the emissive state has changed with increasing doping concentration, as evidenced by the red shift in the emission maximum of Y6 from 800 nm to 925 nm. Additionally, the emission profile and emission energy of Y6-SMAs and Y6-PAs are completely different, as shown in Figure S17, indicating that the emission of these compounds originates from different excited states. Therefore, comparing the emission lifetimes of these two distinctive emissive states is not an appropriate approach, as they belong to two completely different classes of molecules. Basically, they are totally two different classes of molecules without any point to compare.

Author response:

Thanks for the comment. As mentioned above, we believe the differences in PL of Y6-PAs and Y6-SMAs comes from the polymerization, i.e., intra-chain delocalization. According to our optical and DFT results (Figure S1-S2), the electronic structure of Y6-PAs is different from that of Y6, as well. We agree with the reviewer that the emissive states in Y6-SMA and Y6-PA are indeed different (and we are likely the first to point this out). But the purpose of comparing the emission lifetime in these materials is that they are both used in efficient OPV devices, and

the emissive state lifetime is a very important that determines the charge generation efficiency at D–A interface (as pointed out by Brabec et al.).

6. Please fabricate the device show that the PCE and operation stability are indeed improved after all these characterisation and MD calculation.

Author response:

Thanks for the suggestion. We have fabricated devices and test their stability. The device performance of the four systems is summarized in **Table R3**, and the photostability test under 1 Sun illumination is displayed in **Figure R22**. Devices are fabricated in conventional structure with ITO/PEDOT:PSS/active-layer/PNDIT-F3N/Ag, and the efficiencies are comparable to the reported numbers in literature. As shown in Figure R22, PM6:Y6 system decays faster than the other PM6:Y6-PA systems, which could be attributed to the well-known phenomenon so called “burn-in”. The all-polymer systems showed less “burn-in” than small-molecule system. As described previously, “burn-in” mainly comes from the over-purification of at the interfaces driven by the hypo-miscible nature of the blends (*Nat. Mater.* **2021**, *20*, 525-532). This confirms our conclusion that the high miscibility of PM6:Y6-PA blends could “lock” the interfacial morphology and stabilize the devices by suppressing “burn-in”. We have revised our manuscript and Supplementary Information accordingly to include these new device data and discussions.

Added discussion in main text:

“To study the structure–device property relationship, we fabricate device samples of PM6:Y6 and the three PM6:Y6-PAs blends and monitor the change in their PV performance under continuous solar illumination (see Supplementary Fig. S47 and Table S6). The reduced domain purity that we revealed for the Y6-PA blends is consistent with their faster carrier decay (Fig. 2b) and lower device PCE (~13.2–15.5%) compared to Y6 blend (~16.0%). Nevertheless, these Y6-PA blends are still among the most efficient all-polymer OPV systems to date, and we consider that high intra-chain electron transport in these materials helps to reduce recombination losses by allowing charges to move away from the mixed D–A interface. Engineering of the intra-chain charge transport for both polymer donor and acceptor materials may provide a design pathway to further suppress recombination losses in all-polymer blends. For device stability, it is clearly shown that PM6:Y6 device suffered from a more rapid decay in efficiency upon solar illumination compared to the three PM6:Y6-PA blends. This rapid efficiency drop is attributed to the device “burn-in”, which is often observed in non-fullerene SMA-based blends with hypo-miscible D–A morphology (over-purification) at the interfaces. Thanks to the increased D–A miscibility and interfacial percolation in the Y6-PA blends, these all-polymer blends suffer from reduced “burn-in” losses and thus an improved device stability over their SMA counterpart. We consider that the increased D–A miscibility in Y6-PA-based blends helps to explain the overall improvement in device stability compared to Y6-SMA-based blends (see Supplementary Note 3 for extended discussion). Nevertheless, it should be noted that besides blend morphology stability, other factors such as polymer donor degradation and interfacial defects are also likely to affect the overall device stability. Therefore, in addition to developing strategies to improve morphological stability, future research should also target to overcome these factors in order to achieve highly stable and efficient OPV devices that can meet industry requirements.”

Table R3. Summary of photovoltaic performance of devices based on PM6:Y6, PM6:PY-IT, PM6:PYF-T-o and PM6:PY-V- γ .

Blend	V_{oc} (V)	J_{sc} (mA cm ⁻²)	FF (%)	PCE_{avg} (%)	PCE_{max} (%)
PM6:Y6	0.876 ± 0.003	25.55 ± 0.59	71.5 ± 1.6	16.09 ± 0.69	16.94
PM6:PY-IT	0.932 ± 0.002	22.14 ± 0.26	66.4 ± 0.4	13.77 ± 0.23	14.13
PM6:PYF-T-o	0.918 ± 0.007	23.59 ± 1.08	60.7 ± 2.8	13.19 ± 0.15	13.48
PM6:PY-V- γ	0.914 ± 0.003	25.12 ± 0.41	67.0 ± 1.0	15.52 ± 0.21	15.90

Device performance is averaged from 12 cells.

Figure R22. Photostability of devices based on PM6:Y6, PM6:PY-IT, PM6:PYF-T-o, and PM6:PY-V- γ under 1 Sun illumination.

Reviewer #1 (Remarks to the Author):

All issues raised by reviewers have been well addressed in this revised manuscript, it can meet the publishable standard of Nature Communication at present stage. Therefore, I recommend the acceptance without further revisions.

Reviewer #2 (Remarks to the Author):

The authors have adequately addressed all points and presented additional research works fully confirming their original conclusions. In my opinion, the manuscript can be published without further modifications.

Reviewer #3 (Remarks to the Author):

The authors have addressed the raised issues properly and the quality of the manuscript meets the requirement of Nat. Commun. I recommend publication in this version.

Additional Comments:

(1) In Figure S19, the Normalized PL spectrum of PY-IT (0.001 mg/ml) in CB solution is missing.

(2) Based on the Figure S21. Temperature-dependent UV-vis absorption spectra of Y6, PY-IT, PYF-T-o and PY-V- γ in CB solutions (0.005 mg/mL), do show different extend of ground state aggregation. These result could lead to the further modification of the monomer is needed for further improving the PCE and device stability of photovoltaic performance of devices based on, PM6:PY-IT, PM6:PYF-T-o and PM6:PY-V- γ .

REVIEWERS' COMMENTS

Reviewer #1 (Remarks to the Author):

All issues raised by reviewers have been well addressed in this revised manuscript, it can meet the publishable standard of *Nature Communication* at present stage. Therefore, I recommend the acceptance without further revisions.

Author response:

Thanks again for the reviewer's comments and suggestions.

Reviewer #2 (Remarks to the Author):

The authors have adequately addressed all points and presented additional research works fully confirming their original conclusions. In my opinion, the manuscript can be published without further modifications.

Author response:

Thanks again for the reviewer's comments and suggestions.

Reviewer #3 (Remarks to the Author):

The authors have addressed the raised issues properly and the quality of the manuscript meets the requirement of *Nat. Commun.* I recommend publication in this version.

Author response:

Thanks again for the reviewer's comments and suggestions.

Additional Comments:

(1) In Figure S19, the Normalized PL spectrum of PY-IT (0.001 mg/ml) in CB solution is missing.

Author response:

Thanks for pointing this out. We have corrected Figure S19 in the revised SI. The normalized PL spectrum of PY-IT (0.001 mg/mL) is almost identical to that of 0.005 mg/mL, and the large part of them are overlapped.

(2) Based on the Figure S21. Temperature-dependent UV-vis absorption spectra of Y6, PY-IT, PYF-T-o and PY-V- γ in CB solutions (0.005 mg/mL), do show different extend of ground state aggregation. These result could lead to the further modification of the monomer is needed for further improving the PCE and device stability of photovoltaic performance of devices based on, PM6:PY-IT, PM6:PYF-T-o and PM6:PY-V- γ .

Author response:

Thanks for the reviewer's comments. We have added some sentences in the revised SI to discuss this.

"At higher temperatures, the absorption spectra of solutions samples get broadened due to the stronger electron-phonon couplings and likely molecular relaxation. It is observed that there is slight blueshifts for the solutions as the temperature ramps up, while the shifts are rather small, especially for Y6-PAs. These results therefore indicate that with such low concentration, the degree of aggregation of Y6-PAs in solution is negligible."